# Incorporating Industrial By-Products into Geopolymer Mortar: Effects on Strength and Durability

**DOI:** 10.3390/ma16124406

**Published:** 2023-06-15

**Authors:** Tang Van Lam, May Huu Nguyen

**Affiliations:** 1Faculty of Civil Engineering, Hanoi University of Mining and Geology, 18 Pho Vien, Duc Thang, Hanoi 100000, Vietnam; lamvantang@gmail.com; 2Department of Bridge and Tunnel, Faculty of Civil Engineering, University of Transport Technology, 54 Trieu Khuc, Thanh Xuan, Hanoi 100000, Vietnam; 3Civil and Environmental Engineering Program, Graduate School of Advanced Science and Engineering, Hiroshima University, 1-4-1, Kagamiyama, Higashi-Hiroshima, Hiroshima 739-8527, Japan

**Keywords:** granulated blast furnace slag, fly ash, workability, compressive strength, geopolymer mortar, sustainable development

## Abstract

In recent years, the reuse of industrial waste has become increasingly important for sustainable development. Therefore, this study investigated the application of granulated blast furnace slag (GBFS) as a cementitious replacement material in fly-ash-based geopolymer mortar containing silica fume (GMS). The performance changes in the GMS samples manufactured with different GBFS ratios (0–50 wt%) and alkaline activators were evaluated. The results indicated that GBFS replacement from 0 wt% to 50 wt% significantly affects GMS performance, including improving the bulk density from 2235 kg/m^3^ to 2324 kg/m^3^, flexural-compressive strength from 5.83 MPa to 7.29 MPa and 63.5 MPa to 80.2 MPa, respectively; a decrease in water absorption and chloride penetration, and an improvement in the corrosion resistance of GMS samples. The GMS mixture containing 50 wt% GBFS demonstrated the best performances with notable results regarding strength and durability. Owing to the increased production of C-S-H gel, the microstructure of the GMS sample containing more GBFS was denser, as obtained via the scanning electron micrograph analysis results. Incorporating the three industrial by-products into geopolymer mortars was verified when all samples were determined to be in accordance with the relevant Vietnamese standards. The results demonstrate a promising method to manufacture geopolymer mortars that aid sustainable development.

## 1. Introduction

Concrete is a widely used material because of its advantageous structural qualities [1,2,3]. It is used in numerous constructed objects, such as buildings, bridges, highways, and dams. According to the United States Geological Survey, approximately 4.2 billion tons of ordinary Portland Cement (OPC) were produced globally in 2021, with an expected increase to 4.4 billion tons by the end of 2022 [4]. However, this material poses severe environmental issues related to pollution that precedes global warming. For instance, substantial temperatures—approximately 1400 °C—are necessary for cement production with high energy dispersion and emissions. It was reported that cement production (i.e., OPC) accounts for 5–9.5 percent of all CO_2_ emissions [5]. Thus, there is a pressing need to identify an environmentally preferable alternative to conventional concrete to decrease CO_2_ emissions [6,7].

Recent breakthroughs in materials sciences have centred on creating geopolymer concrete as an alternative to traditional concrete [8,9,10]. In theory, geopolymers are the result of alkali activation of any aluminosilicate substance. They have a three-dimensional aluminosilicate network structure and binding properties comparable to those of standard Portland cement (OPC). Compared to OPC, geopolymer concrete (GPC) performs more sustainably and robustly [8,11,12]. The development of GPC offers two key benefits. The first is the decline in the demand for OPC and other concrete types, which would substantially reduce CO_2_ emissions while preserving the performance (i.e., high tensile strength and better thermal insulation properties) [11,13]. The second advantage is the potential use of easily available industrial/agricultural by-products and building wastes, such as granulated blast furnace slag (GBFS) and fly ash (FA), which contribute to landfill waste [14,15].

Incorporating FA into geopolymer concrete offers a novel, beneficial, and environmentally friendly solution for waste material use while minimising the associated adverse effects on the environment and ecosystem [12,16]. FA-based geopolymers may be utilised as green cement with efficient use of natural resources because they often exhibit mechanical strength and durability that are almost on par with hydrated Portland cement [17,18,19]. However, incorporating FA into geopolymer concrete has a main disadvantage related to the low reactivity of FA-based geopolymers [20]. The low-reactivity process leads to slow setting and strength development. In addition, fly ash’s mullite and quartz contents remain unreacted during the process and are dissolved in an alkali-activator solution [21,22].

To address these problems, two different approaches have been investigated. The first uses the mechanical processing of FA, such as fine grinding and mechanical activation. Prior research has been conducted to determine how the mechanical processing of FA affects the reactivity and polymerisation of the substance [23,24]. Incorporating GBFS into FA-based geopolymers can happen differently depending on curing temperatures. At normal temperatures, the reaction is dominated by the dissolution and precipitation of CaO-SiO_2_-H_2_O (C–S–H) gel due to the alkali activation of GBFS. There is only a small interaction of FA and GBFS, probably due to the different kinetics of the dissolution process and species distribution. The improvement in setting time and compressive strength can be explained by forming cementitious C–S–H gel, which improved the setting and hardening of the geopolymer [25,26]. The geopolymerisation at high temperatures (e.g., 60 °C) is dominated by the combined interaction of fly ash and GBFS. This interaction is substantiated by the coexistence of C–S–H and Al_2_O_3_-SiO_2_-H_2_O (A–S–H) gel in the reaction products. The improvement in compressive strength with slag addition may be attributed to the formation of gel phases (C–S–H and A–S–H) and the compactness of microstructure [14,27,28]. In other words, the combination of GBFS and silica fume (SF) under high temperatures could improve the reactivity and polymerisation of FA [27,29,30] and is the main approach of the present study.

When applied to alkali-activated materials, GBFS acts as a precursor for calcium-rich aluminosilicates, enhancing properties such as compressive strength, density, and sulphate resistance [31,32]. Its inclusion in cement can reduce porosity and improve corrosion resistance. Studies have shown that GBFS positively affects sulphate resistance in concrete by reducing 3CaO·Al_2_O_3_ concentration and soluble Ca(OH)_2_, with the concentration of GBFS and sulphate resistance being positively correlated [33,34]. SF is a by-product of ferrosilicon production. It has an extremely fine particle size and is mostly used as a filler. It is a highly reactive pozzolanic material because it contains significant amorphous silicon dioxide. In addition to increasing the strength, the use of SF improves the cohesion between the binder and aggregates, owing to its tiny particle size. Thus, this reduces the mixture’s bleeding rate and segregation potential in its fresh form [35,36].

In the past few decades, GBFS and FA have been widely employed as supplemental materials to mitigate the negative environmental impacts of the concrete industry [37,38]. Approximately 450 million tons of FA and 530 million tons of GBFS are manufactured annually globally [37,39]. Between 25 and 65 percent of the FA and GBFS produced were reportedly recycled and reused [34,40]. Vietnam has approximately 20 thermal power plants, producing approximately 67.3 million tons of coal ash annually. The new plants presently being built are predicted to bring the total amount of coal consumed to 171 million tons, generating an enormous amount of FA by 2030 [41]. Each year, approximately 3 million tons of GBFS are produced by blast furnace pig iron companies in Vietnam [42]. Vietnam’s significant yearly FA output has recently been considered a potential commodity supply. Unfortunately, FA and GBFS are used less frequently in Vietnam and are typically disposed of in landfills, resulting in the loss of raw materials and negatively affecting the environment and human health. Thus far, in Vietnam, although several construction products utilising FA and GBFS as their primary binder materials have been reported [7,38,43], the potential to incorporate such materials into geopolymer mixtures containing silica fume has not yet been investigated.

Therefore, this study investigates the performance of a fly ash-based geopolymer mortar utilising the following Vietnamese by-products: FA, GBFS, and SF. The changes in the slump, compressive strength, density, water absorption, chloride resistance, and corrosion resistance of fly ash-based geopolymer mortar containing silica fume (GMS) were investigated. In addition, alkaline activators (i.e., 10 M sodium hydroxide and sodium silicate solutions) and initial heat curing (60 ± 5 °C) were applied to enhance the mortars’ performance. Based on the acquired data, the changes in these qualities and their relationships were subsequently determined and analysed.

## 2. Experimental Program

### 2.1. Experimental Variables

To investigate the feasibility of local GBFS, FA, and SF, the development of the slump flow, flexural strength, compressive strength, density, water absorption, chloride resistance, and corrosion resistance of fly ash-based geopolymer mortar with silica fume (GMS) was tracked (see Table 1).

### 2.2. Materials

The properties of the materials utilised were assessed in the laboratory in accordance with the relevant standards’ prescriptions. Fine aggregates for GMS mixes were Red River quartz sands (QS) (Hanoi-Vietnam) (Figure 1a), which complied with the TCVN 7570:2006 standard [44] with a fineness modulus and specific gravity of 3.0 and 2.65 g/cm^3^, respectively. Alumino-silicate materials (ASM) included class F fly ash (FA) taken at the thermal power plant “Vung Ang-Ha Tinh-Vietnam” (as per ASTM C618 [45]) (Figure 1b); granulated blast furnace slag (GBFS) of the Hoa Phat iron and steel factory “Hoa Phat–Hai Duong–Vietnam” (Figure 1c), and silica fume SF-90 (SF-90) from “Vina Pacific Co., Ltd., Hanoi, Vietnam” (Figure 1d). 

Table 2 displays the physical characteristics and chemical compositions of FA, GBFS, and SF-90 using the Rietveld refinement with XRD [46] and X-ray fluorescence analysis, (Empyrean model manufactured by a PANalytical instrument, Almelo, The Netherlands), respectively. The FA was primarily comprised of SiO_2_ (54.32 wt%), Al_2_O_3_ (25.47 wt%), and CaO (4.65 wt%); the main components of GBFS were SiO_2_ (36.3 wt%), Al_2_O_3_ (17.97 wt%), and CaO (40.1 wt%), while those of SF-90 were SiO_2_ (91.33 wt%) and Al_2_O_3_ (1.24 wt%). Table 2 shows that the glass contents of FA, GBFS, and SF-90 were 43 wt%, 93 wt%, and 95 wt%, respectively. It also indicates that rising GBFS corresponds to rising CaO and overall amorphous phase contents. The relatively high glass contents of SF-90 and GBFS may be expected to positively affect the strength and microstructural characterisation of the geopolymers.

Figure 2 displays the FA, SF-90, and GBFS particle morphologies by using scanning electron microscopy (JEOL JSM-6390LV, Tokyo, Japan). Herein, FA particles were mostly spherical (Figure 2a) and SF-90 had particles of irregular shape with nano-sized grains (Figure 2b), whereas particles of the GBFS were angular and irregular in shape (Figure 2c). In agreement with Table 2, SEM indicated that the grain size of SF-90 was significantly smaller than that of FA and GBFS.

The X-ray diffraction (XRD) patterns of FA, GBFS, and SF-90 produced with a “D2-PHASER” X-ray diffractometer employing Cu-Kα radiation (Tokyo, Japan) are depicted in Figure 3. A non-crystalline phase existed in the GBFS and SF-90 structures, whereas a high content of quartz and mullite was observed in FA.

In addition, the alkali-activator solution was a mixture of Na_2_SiO_3_ and NaOH solutions. Sodium hydroxide solids with 98.5% purity (Viet-Nhat Co., Ltd., Hanoi, Vietnam) were produced by dissolving the NaOH flakes in water to the desired molarity. The sodium hydroxide solution had a molarity of 10 M and a specific gravity of 1.40 g/cm^3^. The composition of the 10 M NaOH solution was 31.4 wt% and water 68.8 wt%. Sodium silicate was used in the liquid form. The Na_2_SiO_3_ (Viet-Nhat Co., Ltd.) has a specific gravity of 1.55 g/cm^3^, a molar ratio (SiO_2_/Na_2_O) of 2.5, and compositions of 29.50 wt% SiO_2_, 11.80 wt% Na_2_O, and 58.70 wt% H_2_O. The chemical and physical properties of the sodium silicate solution are presented in Appendix A.

The SR-5000F “SilkRoad” (Hanoi-Korea Co., Ltd., Hanoi, Vietnam) was used as the superplasticiser admixture (SP). The specific gravity of the SP was 1.12 g/cm^3^, confirmed by the Vietnamese standard TCVN 8826:2011 [47]. According to the Vietnamese standard TCVN 4506:2012, tap water (W) was utilised to prepare sodium hydroxide solution and cure FA alkali-activated mortar specimens [48]. Herein, the presence of SR-5000F and the minimum amount of water were designed to mitigate the flash setting problem and increase the workability of the FA-based geopolymer mortars containing GBFS [27,49,50]. The steel reinforcement for corrosion resistance was normal Vietnamese steel bars (Thainguyen Tisco Co., Ltd., Hanoi, Vietnam) with a diameter of 10 mm and modulus of elasticity of 20 (GPa).

### 2.3. Mixture Proportions

According to Li et al. [51] and the absolute volume technique, the compositions of the FA alkali-activated mortar mixtures were estimated [52]. The fly ash-based geopolymer mortars were made specifically according to the following principles: the sodium silicate solution to sodium hydroxide solution ratio of 2.0 for all mixes, as referred to in a prior study, and the liquid-to-alumino-silicate materials (L/ASM) ratio of 0.35 by mass (ASM = FA + SF-90 + GBFS) [51,53,54]. The sodium hydroxide solution was first prepared by combining water and 98.5% pure NaOH flakes. The utilised sodium hydroxide solution had a 10M molarity. Sodium silicate was then added to this solution, which was allowed to cool to ambient temperature for 24 h.

To examine the effect of GBFS on the performance of GMS, it was used to replace 0%, 10%, 20%, 30%, 40%, and 50% of FA by weight to make GMS samples named GMS00, GMS10, GMS20, GMS30, GMS40, and GMS50, respectively. Silica fume was used as the modifier and was equal to 10% by weight of ASM. The relative volume of entrapped air in 1 m^3^ of mortar was 2.0% [31,32,55]. For all fly ash alkali-activated mortar mixes, a quartz sand to alumino-silicate material ratio (QS/ASM) of 1.30 was maintained [27,30]. The superplasticiser SR-5000F SilkRoad included 1.0 wt% by weight of ASM [13,14,53]. Details of the GMS mixture proportions used in this study are listed in Table 3.

Notably, the water included in the NaOH 10M and Na_2_SiO_3_ solutions was used to calculate the mix percentage (Table 3) (no extra water was added to the mixtures).

The SiO_2_/Al_2_O_3_, CaO, SiO_2_, and glass contents in the GMS samples are displayed in Appendix B. Here, the individual contributions from FA, SF-90, and GBFS were also used to indicate the overall glass content. The SiO_2_/Al_2_O_3_ ratio was maintained in the vicinity of 2.5, with an increase in GBFS quantity from 0 wt% to 50 wt%, which is characteristic of the structure of geopolymer materials [32]. Furthermore, an increase in the GBFS concentration corresponds to an increase in CaO and the overall glass content.

### 2.4. Samples Preparation

The GMS samples were prepared in the following two processes: first, the raw components were dry-mixed until homogenous mixtures were achieved; next, an alkali-activator solution was added to the dry powder and mixed for 3–5 min. The workability of the mortar mixture was assessed and placed into the corresponding molds. The beams (40 × 40 × 160 mm) were then cast for compressive and flexural strength and water absorption tests, while cylindrical samples (100 × 200 mm) were cast for chloride penetration and accelerated corrosion tests.

To promote the combined interaction of FA and GBFS, the molded samples were sealed in airtight bags and placed in a hot oven at 60 ± 5 °C and at 70% relative humidity for six hours [56,57]. After 24 h of heat curing, all the samples were demoded and subjected to normal maintenance (temperature of 25 ± 2 °C and humidity of approximately 90%) until testing. Herein, the two-stage curing cycle was used to avoid the overlapping of the dissolution–precipitation reaction at ambient temperature (25 ± 2 °C) and geopolymerisation reactions at heat curing (60 ± 5 °C).

### 2.5. Test Methods

The GMS samples underwent tests for slump flow, compressive strength, density, water absorption, chloride resistance, and corrosion resistance in line with the appropriate Vietnamese requirements to examine engineering qualities and long-term performance. The test results at each age were determined by calculating the arithmetic means for the three GMS samples. The microstructural properties of the materials were also examined (SEM model QUANTA 450, Tokyo, Japan). The following provides a detailed description of each test method.

#### 2.5.1. Slump Flow and Bulk Density

The slump flow of the GMS samples was examined immediately and 15 and 30 min after mixing to check the workability. A mini-slump cone with dimensions of 100 × 70 × 60 mm was used in accordance with TCVN 3121-3:2003 [58]. Fresh GMS mortar was poured into a steel plate slump cone. The slump cone was elevated vertically, and the mortar flowed out freely. The maximum and orthogonal diameters were determined, and the mean of the three diameters was reported as the final slump flow value.

The bulk density ($\rho_{GMS}$) was calculated via its weight (i.e., with a 1000 mL container) using Equation (1) [59].
(1)$$\rho_{GMS}=\frac{m_{GMS}}{1\times 10^{-3}}$$
where $\rho_{GMS}$ denotes the bulk density of fresh GMS, kg/m^3^; $m_{GMS}$ is the weight of fresh mortar (kg).

#### 2.5.2. Compressive and Flexural Strength

According to TCVN 6016:2011, compressive tests were performed on beam samples (40 × 40 × 160 mm) [60]. Using a 500T computer-controlled compression test (ADVANTEST9-Control—Italy), the experiment was conducted at 1, 3, 7, 14, 28, 56, and 90 d with a constant loading rate of 2400 ± 200 N/s. The flexural strength of the GMS was measured using beam specimens (40 × 40 × 160 mm) with a loading rate of 50 ± 10 N/s, as per the standard TCVN 6016:2011. The test was performed after 28 d of curing, and the average flexural strength of the three specimens was reported.

#### 2.5.3. Water Absorption

The water absorption of the GMS specimens was determined according to TCVN 3113:1993 at an age of 28 d [61]. The test procedures included the following: (1) measuring the dry weight (*m*_0_) after oven drying to a constant weight; (2) immersing in water for 24 h at 27 ± 2 °C; (3) checking the saturated weight (*m*_1_); and (4) calculating the water absorption using Equation (2).
(2)$$W\left(\%\right)=\left(\frac{m_{1}-m_{0}}{m_{0}}\right)\times 100\%$$

#### 2.5.4. Chloride Penetration

According to TCVN 9337:2012, the permeability of chloride ions in the structure of GMS samples was assessed [62] at 28 d of curing age. This technique measures the amount of electrical energy that passes through a cylindrical GMS sample with a diameter of 100 mm and a height of 50 mm over the course of 6 h. In addition, the total electrical energy that passed through the GMS sample under test was estimated in Coulombs (C).

#### 2.5.5. Accelerated Corrosion Test

The test procedures were in accordance with NT BUILD 356 [63]. The approach under consideration aims for an expedited evaluation of the degree of protection against corrosion in mortar constructions operating in demanding settings 28 d after curing. Figure 4 depicts the experimental apparatus and GMS samples with which the experiment was performed. The samples had the shape of cylinders with dimensions of 100 × 200 mm, and a steel rod 10 mm in diameter was placed in the centre of each sample. In this investigation, the subjects were exposed to a 3% NaCl solution for 90 d.

## 3. Results and Discussion

### 3.1. Slump Flow and Bulk Density of Fresh Mortars

Slump flow is a reliable index reflecting the workability and other properties of fresh and hardened geopolymer mortars [64]. Figure 5 shows the average value of the slump flow of the GMS samples. All GMS samples generally had excellent workability, with an 18–21 cm slump meeting Vietnamese construction requirements. The influence of GBFS content on the slump flow values was observed, with a lower slump flow obtained at a higher level of GBFS content. When the amount of GBFS increased, the proportion of particles with angular and irregular shapes (see Figure 3c) increased, whereas the proportion of spherical particles (Figure 3a) in FA decreased. The higher presence of angular particles and irregular shapes reduces the slump flow values [64,65]. In addition, because GBFS has a smaller particle size, more water would be absorbed as more GBFS was included, resulting in lower slump flow [66].

The influence of GBFS on the bulk density of fresh GMS mortars is shown in Figure 6. The results indicated that adding GBFS at different levels increased the average density of the mortar mixtures from 2235 kg/m^3^ to 2324 kg/m^3^. The bulk density values of fresh mortars with 0%, 10%, 20%, 30%, 40%, and 50% by weight of GBFS were 2235, 2253, 2270, 2288, 2305, and 2324 kg/m^3^, respectively, which were 18, 35, 53, 70 and 89 kg/m^3^ higher than that of the GMS00 (0 wt% GFBS), respectively. The higher specific gravity of the GFBS (see Table 2) may explain the increment in the bulk density values. In addition, the smaller particle size of GBFS compared to that of FA could be another reason. Accordingly, with a higher amount of GFBS in the mixture, a better packing density was obtained, which increased the bulk density [43,67].

### 3.2. SEM Observation

SEM analysis was used to describe the changes in the morphology of the microstructure of the GMS samples at 7 d. Figure 7 depicts the changes in the microstructure of the GMS samples with different amounts of GFBS. Owing to the increased production of C-S-H gel, the microstructure of the GMS sample containing more GBFS was generally denser. In the GMS00 samples, partially reacted cenospheres were the primary characteristic (Figure 7a). In the GMS10 and GMS20 samples with 10% and 20% by weight of GBFS, respectively, the cenosphere was coated with the reaction product (Figure 7b,c). The structure of the gel phase at the surface of the FA particles, which grows outwards in the hydroxide system, was observed in agreement with a previous finding [68]. When the GBFS content increased to 30%, 40%, and 50%, small prismatic structures with diffused borders and long fibers in clusters were observed (Figure 7d,e), and fibrous products were detected (Figure 7f).

Notably, the highly active nano-SiO_2_ content of SF-90 tends to react rapidly with the Na_2_O content of the alkaline medium and the CaO concentration of GBFS. It improves the microstructure of geopolymer mortars by generating CaO-Al_2_O_3_-SiO_2_-H_2_O, CaO-SiO_2_-H_2_O, and Al_2_O_3_-SiO_2_-H_2_O gels. Hence, owing to these gels’ enhanced production, the GMS sample’s microstructure with a greater GBFS concentration was denser [27,30]. The results provide reliable proof to clarify the changes in compressive strength, flexural strength, density, water absorption, chloride resistance, and corrosion resistance, as presented in the following sections.

### 3.3. Compressive and Flexural Strength

The compressive strength is critical as the primary mechanical index used to evaluate the GMS performance [64,69]. The changes in the compressive strength of the GMS samples produced with different GBFS contents are indicated in Figure 8. Over time, the overall compressive strength of the samples increased. The increase in compressive strength may result from the formation of extra hydration products, which fill the porous microstructure of the geopolymer mortar [64,65]. As a result, the number of hydration products increases as the hydration stages develop, leading to a more compact microstructure and a higher compressive strength. The results at 28 d of curing age indicate that the 50 wt% GMS samples had the highest compressive strength, followed by the (10–40) wt% GMS samples. The compressive strengths of the GMS samples with 50%, 40%, 30%, 20%, and 10% by weight of GBFS were 80.2, 78.2, 76.4, 72.8, and 69.7 MPa, respectively, which were 26.30%, 23.15%, 20.31%, 14.65%, and 9.76% greater than that of the GMS00 (0 wt% GFBS), respectively. Based on the results at 28 d, the GMS samples can be classified as M30 under TCVN 4314:2003 [70].

Furthermore, the influence of GBFS on the compressive strength of the GMS samples was determined, with a positive correlation between compressive strength and GBFS content at each time point. For example, the results at 90 d indicate a decrease in the compressive strength from GMS50 (50 wt% GBFS) to GMS00 (0 wt% GBFS). Specifically, the compressive strengths of the GMS samples containing 50%, 40%, 30%, 20%, and 10% by weight of GBFS were 98.4, 96.1, 93.9, 89.1, and 85.5 MPa, respectively, which were 26.15%, 23.21%, 20.38%, 14.23%, and 9.62% higher than that of GMS00 (0 wt% GFBS), respectively. Herein, the increment in compressive strength could be supported by the denser bulk density of the GMS samples produced with GBFS (Figure 6) and its lower porosity [43,64]. Moreover, the SEM observations (Figure 7) and the strong positive relationship between compressive strength and bulk density (Figure 9) provide compelling evidence for explaining the changes in compressive strength.

Table 4 shows the changes in the compressive and flexural strength values at 1, 3, 7, 14, 28, 56, and 90 d, along with the changes in SiO_2_/Al_2_O_3_ and glass content. The experimental results proved that GBFS and glass content positively correlate with compressive and flexural strength increments. More specifically, when the percentage of glass increased from 48.2 wt% to 73.2 wt% (see Table 4), the hydration products steadily increased, resulting in a denser microstructure [64,71]. In addition, regarding reactivity, the high CaO and Al_2_O_3_ concentrations of GBFS paired with the high SiO_2_ content of SF-90 are essential for triggering the reaction with the activator to produce calcium aluminum silicate hydrate products [20]. The flexural-to-compressive strength ratios of the test samples fluctuated only from 1/11 to 1/9, similar to that of traditional cement mortars [72].

### 3.4. Water Absorption

The water absorption of geopolymer mortar depends on its composition, curing circumstances, and aggregate porosity, and is a helpful index to evaluate its durability [73,74]. Owing to their decreased porosity and chemical resistance, geopolymer mortars absorb less water than Portland cement mortars [64]. The changes in the water absorption of the GMS samples at 28 d are shown in Figure 10, with the lower water absorption obtained at a higher amount of GBFS. For instance, at 28 d, the water absorptions of the GMS specimens with 0%, 10%, 20%, 30%, 40%, and 50% by weight of GBFS were 7.15%, 6.48%, 5.94%, 5.67%, 4.86%, and 4.15%, respectively. The lower water absorption could be closely related to the bulk density (Figure 6) and porosity [43,64]. All geopolymerisation products of these compounds filled the gaps in the binder stone because of the geopolymerisation reaction of aluminosilicate materials, such as FA, GBFS, and SF-90, resulting in a denser microstructure and reduced water absorption of the GMS specimens. In addition, SEM results (Figure 7) and the close correlation between water absorption and bulk density (Figure 11) give solid evidence for understanding the variations in water absorption.

In addition, with the same liquid-to-alumino-silicate material ratio of 0.35, the water absorption of all GMS produced with GBFS, that is, from 10 wt% to 50 wt% GBFS, was lower than that of the control specimens GMS00 (see Figure 9). Because GBFS is glassy, the reaction product of the system depends on the response of its Al_2_O_3_ and CaO concentrations, as well as its SiO_2_ content. Thus, higher levels of GBFS are associated with increased amounts of C-A-S-H and N-C-S-H reaction products.

### 3.5. Chloride Resistance

An important aspect of concrete longevity is its resistance to chloride attack. The quick chloride permeability test is frequently used to determine the chloride resistance in concrete (RCPT). Figure 12 displays the changes in the total charge passed (Coulomb) for the GMS samples at the 28 d curing age. The results at 28 d showed that GMS50 had the lowest charge passing value, followed by GMS40 and GMS00, in increasing order. Based on TCVN 9337:2012 [62], the GMS samples were classified as very low (GMS50), low (GMS40, GMS30, and GMS20), or moderate (GMS10 and GMS00). This finding demonstrates that the GMS sample is resistant to chloride penetration, providing a new solution for the construction sector.

In addition, Figure 12 shows how GBFS affected the total charge passed. The total charge passed was demonstrated to inversely correlate with GBFS concentration when SF-90 was held constant, which is consistent with variations in water absorption (see Figure 8). Specifically, the charge passed values of the GMS samples with 50%, 40%, 30%, 20%, and 10% by weight of GBFS were 842C, 985C, 1437C, 1680C, and 2180C, respectively. This is compared to GMS00 (0 wt% GFBS), which had charge pass values that were 71.42%, 66.56%, 51.21%, 41.97%, and 25.97% lower. The lower charge passed values of the GMS samples could be explained by the higher bulk density (Figure 6) and lower porosity of the GMS samples produced with GBFS [43,64]. In addition, the SEM results (Figure 7) and the high correlation between the total charge passed, and bulk density (Figure 13) provide robust evidence for explaining the chloride resistance variations.

### 3.6. The Time Initial Crack

Figure 14 depicts the beginning fracture of the test samples for six compositions to evaluate the corrosion of reinforcement in the environment of 3% NaCl solution by the NT Build 356 standard [63]. The results showed that for all tested GMS with different GBFS contents, the experimental time for the destruction of the tested samples was 45–88 d (see Figure 15). In detail, mortar combinations containing GBFS exhibited significantly longer destruction durations compared to the control mixture, with the order decreasing from GMS50 to GMS00. (GMS00). Herein, the denser microstructure (Figure 7), increased bulk density (Figure 16a), and decreased water absorption (Figure 16b) of GMS samples containing GBFS might all be contributing factors to the longer time initial crack.

## 4. Conclusions

This study examined the performance of a fly ash-based geopolymer mortar using the following three Vietnamese by-products: FA, GBFS, and SF. Changes in the slump flow, compressive strength, density, water absorption, chloride resistance, and corrosion resistance of the GMS samples were explored in detail. The following conclusions were reached in light of the experimental findings:The Vietnamese GBFS content significantly affected the geopolymer mortars’ workability mixtures and strength behaviour. Increasing the GBFS content from 0 wt% to 50 wt% reduced the slump of the mixtures significantly, whereas it had less effect on the densities of the GMS samples. The slump and density of the GMS samples were recorded in the range of 18–21 cm and 2235–2324 kg/m^3^, respectively.The compressive strength of the GMS samples increased from 1 d to 90 d due to the curing period test; the flexural-to-compressive strength ratios of the test samples only changed from 1/11 to 1/9. The GBFS concentration significantly affected the flexural-compressive strength values at 28 d of curing, which ranged from 5.83 MPa to 7.29 MPa and 63.5 MPa to 80.2 MPa, respectively.The GBFS replacement amount reduced the water absorption and chloride penetration while enhancing the corrosion resistance of the GMS samples. This is because all geopolymerisation products of FA, SF-90, and GBFS with the alkali-activator solution filled the gaps in the binder stone, hence decreasing the water absorption of the GMS-sample. Extremely low chloride ion permeability may be achieved by including 50 wt% GBFS in the mixes. Based on the standard NT Build 356 results, the initial cracking of the test mortar occurred between 45 and 88 d.

The results of this study verified the applicability of incorporating three Vietnamese by-products (FA, GBFS, and SF) to produce geopolymer mortars, as verified by the relevant Vietnamese standards. It also clarified the great potential of GBFS to improve the performance of GMS products, including flexural-compressive strength, corrosion of reinforcement, water absorption, and chloride resistance. Given the constraints of this study, future research should analyse and evaluate the long-term evolution of compressive-flexural strength, water absorption, bulk density, and void volume. Furthermore, further studies should use FTIR and XRD methods to prove the newly created gels in the microstructure of mortars.

## Figures and Tables

**Figure 1 materials-16-04406-f001:**
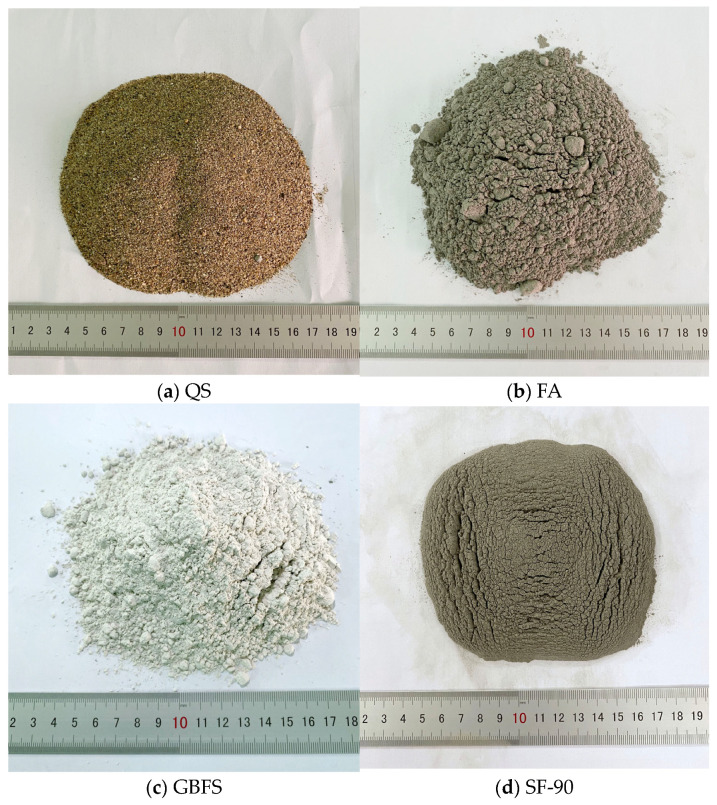
Types of raw materials.

**Figure 2 materials-16-04406-f002:**
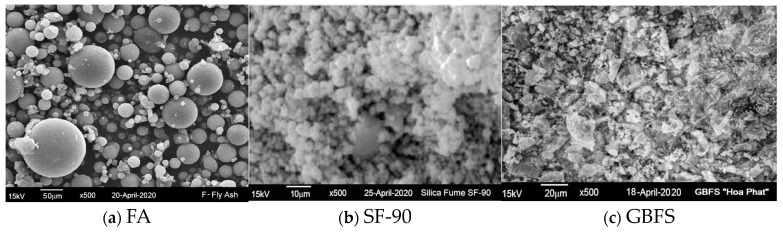
SEM micrographs of (**a**)—FA “Vung Ang”, (**b**)—Silica fume SF-90, and (**c**)—GBFS “Hoa Phat”.

**Figure 3 materials-16-04406-f003:**
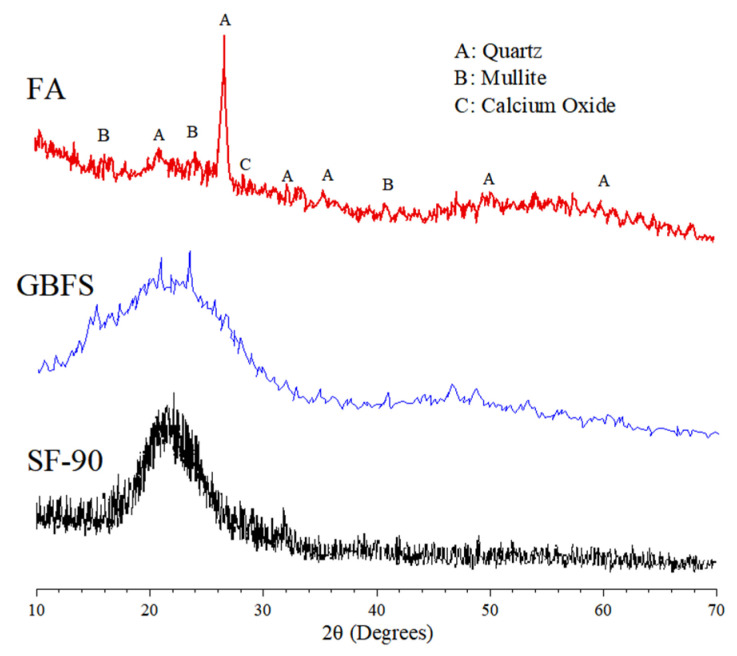
XRD patterns of FA, GBFS, and SF-90.

**Figure 4 materials-16-04406-f004:**
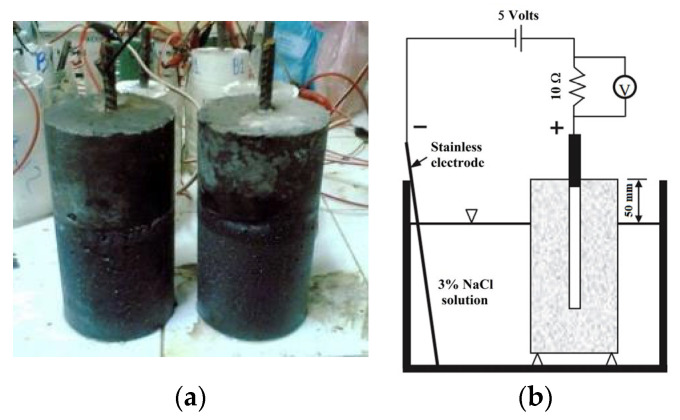
Accelerated corrosion test: (**a**) preparation of the samples and (**b**) scheme test setup.

**Figure 5 materials-16-04406-f005:**
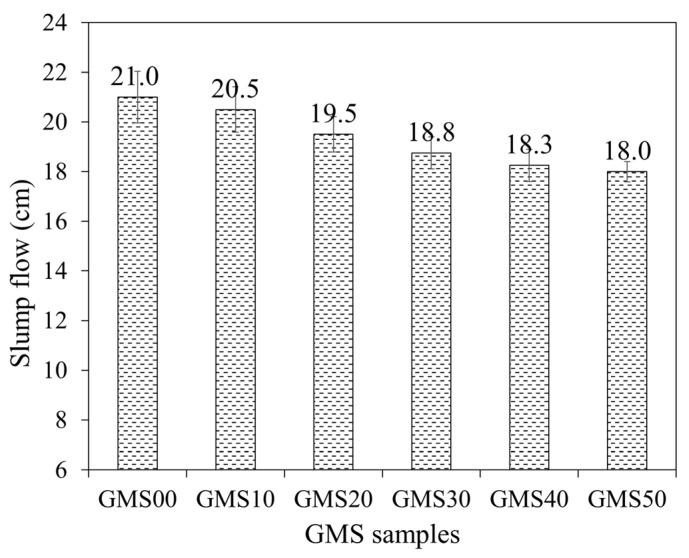
The average value slump cone of GMS fresh mortars.

**Figure 6 materials-16-04406-f006:**
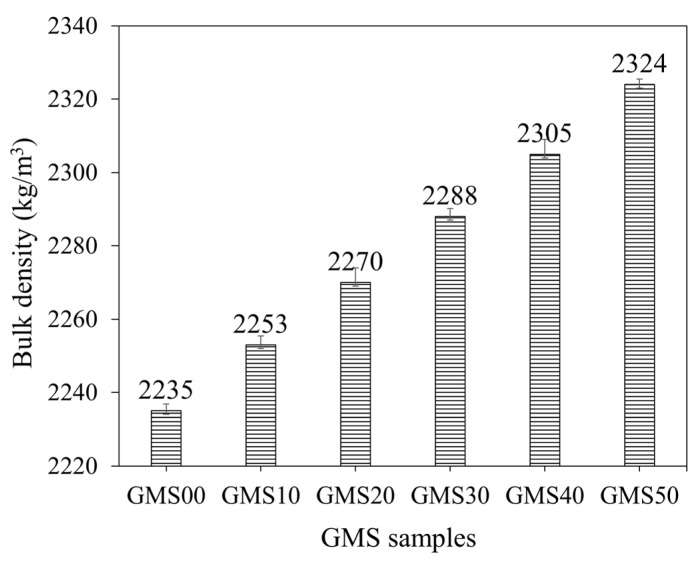
Bulk density of fresh GMS mortars.

**Figure 7 materials-16-04406-f007:**
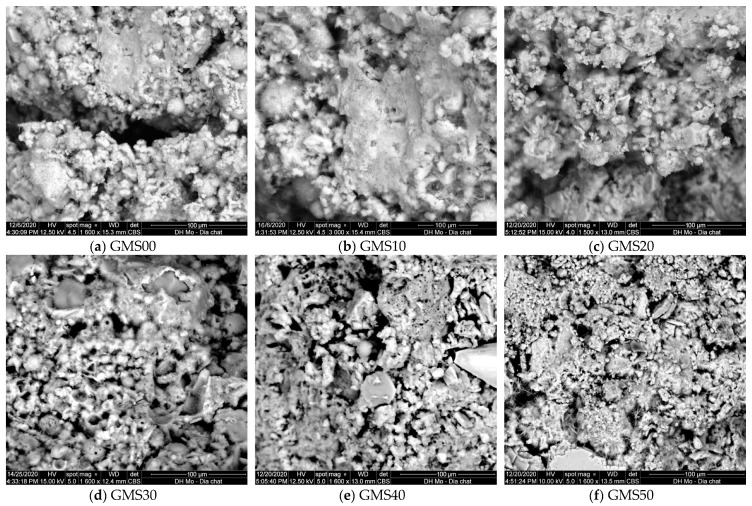
SEM of GMS samples.

**Figure 8 materials-16-04406-f008:**
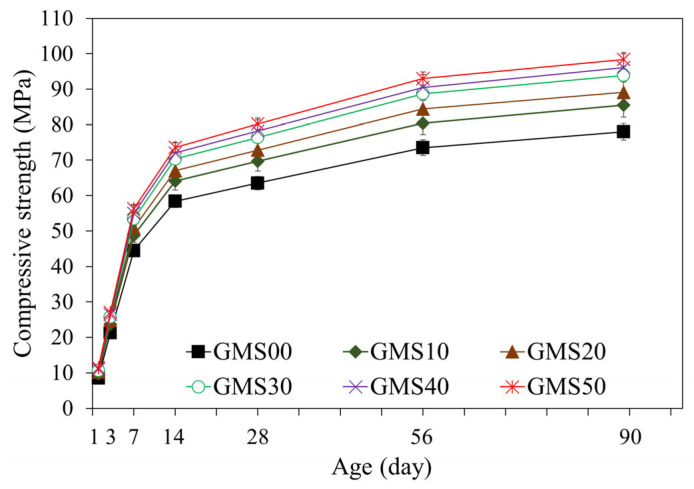
Changes in compressive strength of GMS samples.

**Figure 9 materials-16-04406-f009:**
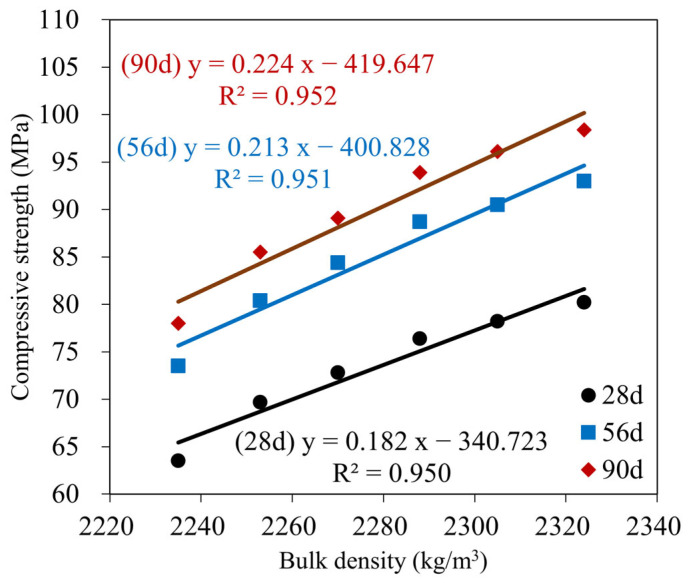
Correlation between compressive strength and bulk density of GMS samples at 28, 56, and 90 d.

**Figure 10 materials-16-04406-f010:**
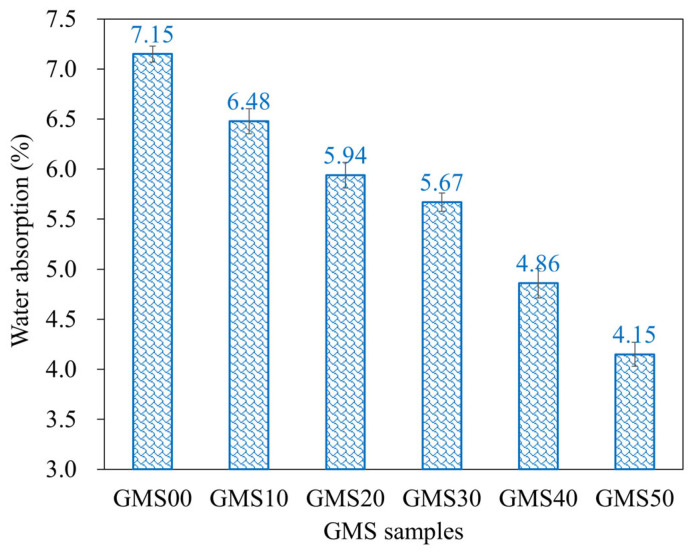
Water absorption of hardened GMS mortars at 28 d.

**Figure 11 materials-16-04406-f011:**
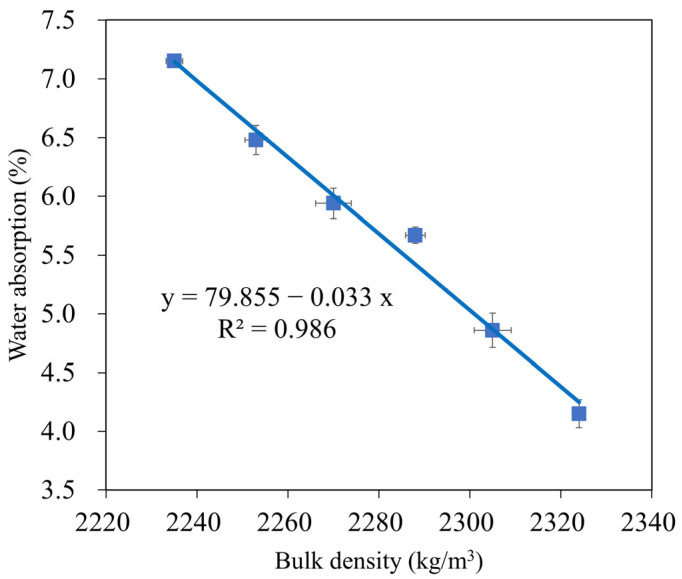
Correlation between water absorption and bulk density of GMS samples.

**Figure 12 materials-16-04406-f012:**
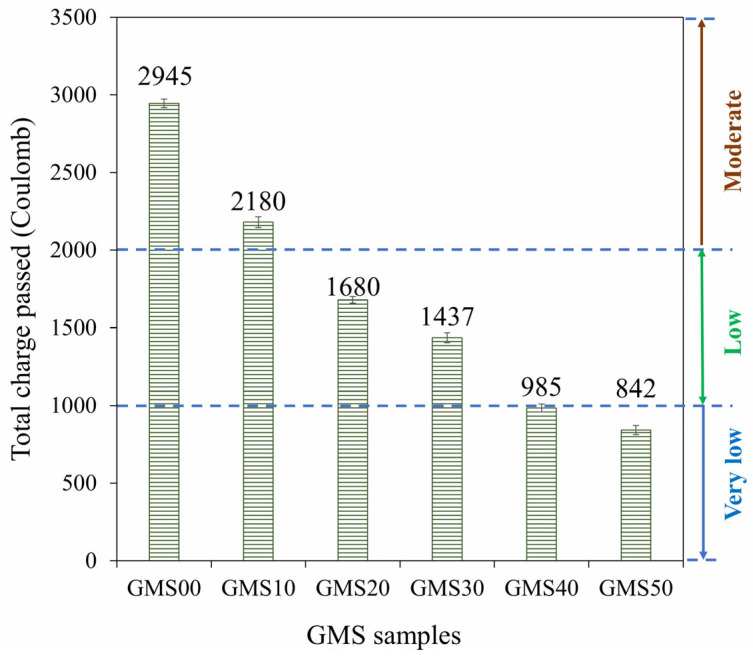
Total charge passed through GMS samples.

**Figure 13 materials-16-04406-f013:**
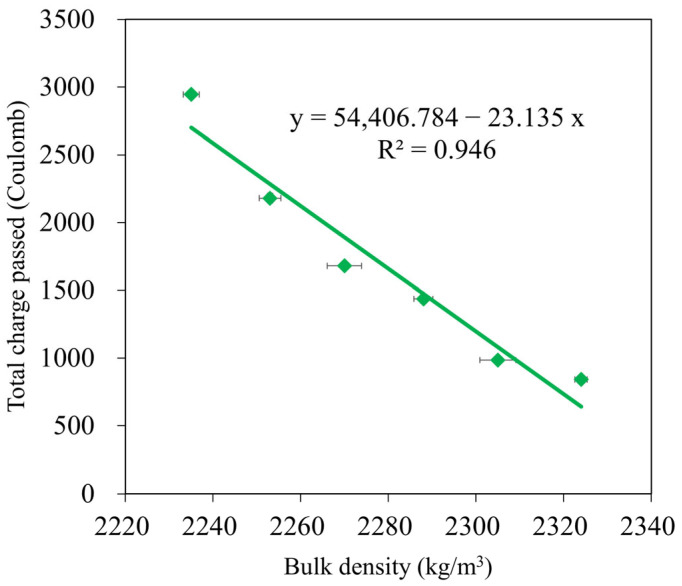
Correlation between total charge passed and bulk density of GMS samples.

**Figure 14 materials-16-04406-f014:**
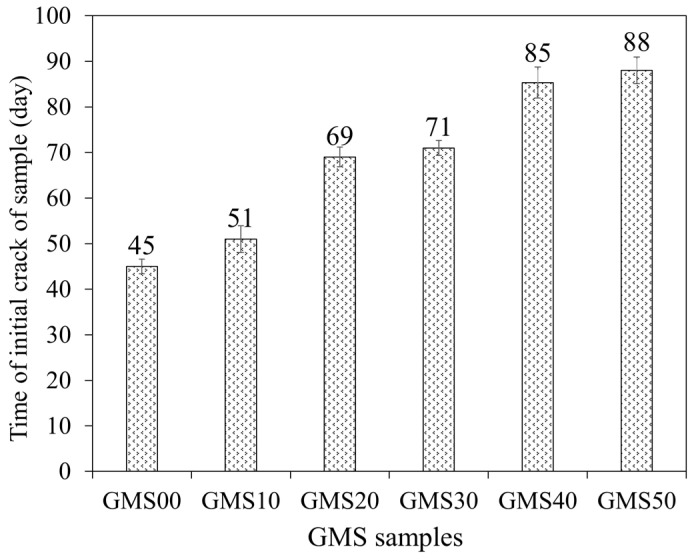
The time initial crack of GMS samples.

**Figure 15 materials-16-04406-f015:**
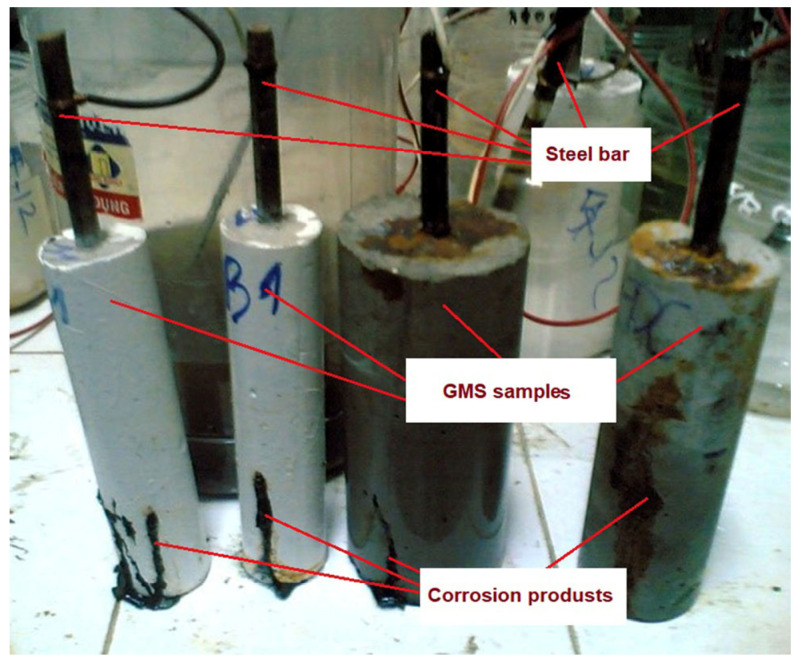
Corrosion failure of GMS samples.

**Figure 16 materials-16-04406-f016:**
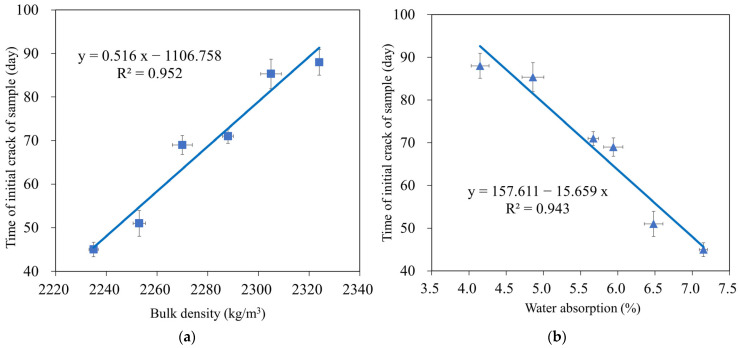
Correlation between time initial crack and (**a**) bulk density and (**b**) water absorption of GMS samples.

**Table 1 materials-16-04406-t001:** Test variables.

Variable	Value/Item
GBFS/(FA + GBFS + SF), (wt%)	0, 10, 20, 30, 40, and 50
Evaluated properties	Slump flow, bulk density, compressive strength, flexural strength, density, water absorption, chloride resistance, and corrosion resistance.

**Table 2 materials-16-04406-t002:** Physical properties and chemical compositions of FA, SF-90, and GBFS.

Items	FA	SF-90	GBFS
Physical properties			
Specific gravity	2.32	2.23	2.92
Specific surface area (m^2^/g)	1.30	18.17	0.92
Mean particle size (μm)	26.6	<1	20.8
Glass content (wt%)	43	95	93
Major crystalline phases	Mullite, Quartz	-	Gehlinite
Average chemical composition (wt%)			
SiO_2_	54.32	91.33	32.96
Al_2_O_3_	25.47	1.24	17.97
Fe_x_O_y_	5.14	0.35	0.86
CaO	4.65	-	36.08
MgO	1.28	-	8.39
Na_2_O	1.12	1.11	-
K_2_O	1.57	1.12	-
SO_3_	1.45	-	0.72
P_2_O_5_	1.25	2.31	0.25
Loss upon ignition	3.75	2.54	2.77

**Table 3 materials-16-04406-t003:** Ingredient proportions of GMS samples.

Mixtures	FA (kg/m^3^)	GBFS (kg/m^3^)	SF-90 (kg/m^3^)	QS (kg/m^3^)	SP (kg/m^3^)	NaOH (kg/m^3^)	Na_2_SiO_3_ (kg/m^3^)	W (kg/m^3^)
GMS00	756	0	84	1092	8.40	30.8	81	182
GMS10	677	85	85	1101	8.47	31.0	81.6	184
GMS20	597	171	85	1109	8.53	31.3	82.2	185
GMS30	516	258	86	1118	8.60	31.5	82.9	187
GMS40	433	347	87	1127	8.67	31.8	83.5	188
GMS50	349	437	87	1136	8.74	32.0	84.2	190

**Table 4 materials-16-04406-t004:** The average value of flexural-compressive strength of GMS samples.

Sample Code	SiO_2_/Al_2_O_3_	Glass Content (wt%)	Flexural Strength (MPa)	Compressive Strength at Different Curing Ages (MPa)						
				1 d	3 d	7 d	14 d	28 d	56 d	90 d
GMS-01	2.56	48.2	5.83	8.5	21.3	44.5	58.4	63.5	73.5	78
GMS-02	2.54	53.2	6.45	9.7	23.7	48.7	64.1	69.7	80.4	85.5
GMS-03	2.53	58.2	6.53	10.2	24.8	50.8	67	72.8	84.4	89.1
GMS-04	2.51	63.2	7.05	10.7	25.7	53.4	70.3	76.4	88.7	93.9
GMS-05	2.49	68.2	7.11	11	26.5	55	72	78.2	90.5	96.1
GMS-06	2.47	73.2	7.29	11.4	27.1	56.3	73.6	80.2	93	98.4

## Data Availability

The data are not publicly available due to the privacy of program data.

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
