# Peer review of "Incorporating Industrial By-Products into Geopolymer Mortar: Effects on Strength and Durability"

_materials, 2023, doi:10.3390/ma16124406_

Round 1
Reviewer 1 Report
The manuscript is interesting, but the authors must improve a few points for publication.
Introduction
It is very interesting but is more focused on the amount of FA and GBFS produced worldwide and in Vietnam. More results obtained from other research are important to show the recent status of investigations and to justify the research significance of work developed by authors (the innovative aspect of the manuscript).
Experimental program
Table 1 - how are the d10 and d90 particle sizes?
Please just avoid presenting commercial names in a technical paper. References regarding manufacturers must be present in the acknowledgment section.
Why the silica fume was used as a modifier? I need help understanding this point.
Results and Discussions
Well explained and discussed accordingly.
In my opinion, the manuscript must be accepted for publication.
Author Response
Reviewer #1:
The manuscript is interesting, but the authors must improve a few points for publication.
Response:
Thank you very much for your positive evaluation.
- Introduction
It is very interesting but is more focused on the amount of FA and GBFS produced worldwide and in Vietnam. More results obtained from other research are important to show the recent status of investigations and to justify the research significance of work developed by authors (the innovative aspect of the manuscript).
Response:
Although there have been several studies on this subject in earlier research [1–3], combining three by-products, FA, GBFS, and SF, into geopolymer mortar in Vietnam has not yet been explored, according to the reviewer's perceptive analysis of this work. The results of the current study thus point to a feasible technique for creating geopolymer mortars that support sustainable development. In other words, this research will make it possible to take a fresh approach to developing sustainable materials and resolving Vietnam's environmental issues. We have mentioned the research significance at the end of the abstract.
In addition, the revised manuscript provided more results obtained from related research in L63-69, page 2.
From:
The second approach utilises modified materials, such as silica fume (SF) and GBFS. Herein, the combination of GBFS and SF could improve the reactivity and polymerisation of FA [25–27] and is the main approach of the present study.
To:
The second approach utilises modified materials, such as silica fume (SF) and GBFS. Incorporating GBFS into FA-based geopolymers can happen differently depending on curing temperatures. At normal temperatures, the reaction is dominated by the dissolution and precipitation of C–S–H gel due to the alkali activation of GBFS. There is only a small interaction of FA and GBFS, probably due to the different kinetics of the dissolution process and species distribution. The improvement in setting time and compressive strength can be explained by forming cementitious C–S–H gel, which improved the setting and hardening of the geopolymer [25,26]. The geopolymerisation at high temperatures (e.g., 60 °C) is dominated by the combined interaction of fly ash and GBFS. This interaction is substantiated by the coexistence of C–S–H and A–S–H gel in the reaction products. The improvement in compressive strength with slag addition may be attributed to the formation of gel phases (C–S–H and A–S–H) and the compactness of microstructure [14,27,28]. In other words, the combination of GBFS and SF under high temperatures could improve the reactivity and polymerisation of FA [27,29,30] and is the main approach of the present study.
For the modifications, we have added the followings references:
[25] Li, Z.; Liu, S. Influence of Slag as Additive on Compressive Strength of Fly Ash-Based Geopolymer. Journal of Materials in Civil Engineering 2007, 19, 470–474, doi:10.1061/(ASCE)0899-1561(2007)19:6(470).
[26] Buchwald, A.; Hilbig, H.; Kaps, Ch. Alkali-Activated Metakaolin-Slag Blends—Performance and Structure in Dependence of Their Composition. J Mater Sci 2007, 42, 3024–3032, doi:10.1007/s10853-006-0525-6.
[27] Singh, B.; Ishwarya, G.; Gupta, M.; Bhattacharyya, S.K. Geopolymer Concrete: A Review of Some Recent Developments. Construction and Building Materials 2015, 85, 78–90, doi:10.1016/j.conbuildmat.2015.03.036.
[28] Zhang, P.; Gao, Z.; Wang, J.; Guo, J.; Hu, S.; Ling, Y. Properties of Fresh and Hardened Fly Ash/Slag Based Geopolymer Concrete: A Review. Journal of Cleaner Production 2020, 270, 122389, doi:10.1016/j.jclepro.2020.122389.
- Experimental program Table 1 - how are the d10 and d90 particle sizes?
Response:
Table 1 describes the test variables, presented first to provide an overview of evaluated properties in the experimental program. Then the detailed experimental program was presented in the following sections (from 2.2 to 2.5). The particle sizes of fly ash, silica fume, and slag were shown in Table 2 in L303-305, page 11. Due to the limitations of experimental equipment, in the present study, the mean particle size was employed instead of d10 and d90.
- Please just avoid presenting commercial names in a technical paper. References regarding manufacturers must be present in the acknowledgment section.
Response:
Based on the comment, we have skipped commercial names in the whole manuscript.
- Why the silica fume was used as a modifier? I need help understanding this point.
Response:
In fact, the silica fume was used as the modifier due to it having high fineness and high glass contents. In detail, the silica fume (SF-90) was primarily comprised of SiO2 (91.33%) in amorphous phase content. With its high fineness, silica fume worked as a microstructural filler of geopolymers mortar. In addition, the relatively high glass contents of SF-90 may be expected to positively affect the reaction of silica fume with fly ash, thereby improving the microstructural characterization and durability of geopolymers mortar.
- Results and Discussions
Well explained and discussed accordingly. In my opinion, the manuscript must be accepted for publication.
Response:
Thank you very much for your keen observations.
Additional revision:
During the improvement of our manuscript based on the comments/suggestions of the reviewers, we found several minor mistakes. Thus, we would like to update them the whole revised manuscript.
References
- Kumar, S.; Kumar, R.; Mehrotra, S.P. Influence of Granulated Blast Furnace Slag on the Reaction, Structure and Properties of Fly Ash Based Geopolymer. J Mater Sci 2010, 45, 607–615, doi:10.1007/s10853-009-3934-5.
- Singh, B.; Ishwarya, G.; Gupta, M.; Bhattacharyya, S.K. Geopolymer Concrete: A Review of Some Recent Developments. Construction and Building Materials 2015, 85, 78–90, doi:10.1016/j.conbuildmat.2015.03.036.
- Zhang, P.; Gao, Z.; Wang, J.; Guo, J.; Hu, S.; Ling, Y. Properties of Fresh and Hardened Fly Ash/Slag Based Geopolymer Concrete: A Review. Journal of Cleaner Production 2020, 270, 122389, doi:10.1016/j.jclepro.2020.122389.
- Wang, Y.; Xiao, R.; Hu, W.; Jiang, X.; Zhang, X.; Huang, B. Effect of Granulated Phosphorus Slag on Physical, Mechanical and Microstructural Characteristics of Class F Fly Ash Based Geopolymer. Construction and Building Materials 2021, 291, 123287, doi:10.1016/j.conbuildmat.2021.123287.
- Oderji, S.Y.; Chen, B.; Shakya, C.; Ahmad, M.R.; Shah, S.F.A. Influence of Superplasticizers and Retarders on the Workability and Strength of One-Part Alkali-Activated Fly Ash/Slag Binders Cured at Room Temperature. Construction and Building Materials 2019, 229, 116891, doi:10.1016/j.conbuildmat.2019.116891.
- Li, Z.; Liu, S. Influence of Slag as Additive on Compressive Strength of Fly Ash-Based Geopolymer. Journal of Materials in Civil Engineering 2007, 19, 470–474, doi:10.1061/(ASCE)0899-1561(2007)19:6(470).
- Buchwald, A.; Hilbig, H.; Kaps, Ch. Alkali-Activated Metakaolin-Slag Blends—Performance and Structure in Dependence of Their Composition. J Mater Sci 2007, 42, 3024–3032, doi:10.1007/s10853-006-0525-6.
- Shadnia, R.; Zhang, L.; Li, P. Experimental Study of Geopolymer Mortar with Incorporated PCM. Construction and Building Materials 2015, 84, 95–102, doi:10.1016/j.conbuildmat.2015.03.066.
- Chithambaram, S.J.; Kumar, S.; Prasad, M.M. Thermo-Mechanical Characteristics of Geopolymer Mortar. Construction and Building Materials 2019, 213, 100–108, doi:10.1016/j.conbuildmat.2019.04.051.
- Prochon, P.; Zhao, Z.; Courard, L.; Piotrowski, T.; Michel, F.; Garbacz, A. Influence of Activators on Mechanical Properties of Modified Fly Ash Based Geopolymer Mortars. Materials 2020, 13, 1033, doi:10.3390/ma13051033.
- Nguyen, M.H.; Nakarai, K.; Nishio, S. Durability Index for Quality Classification of Cover Concrete Based on Water Intentional Spraying Tests. Cement and Concrete Composites 2019, 104, 103355, doi:10.1016/j.cemconcomp.2019.103355.
- Nguyen, M.H.; Nishio, S.; Nakarai, K. Effect of Temperature on Nondestructive Measurements for Air Permeability and Water Sorptivity of Cover Concrete. Construction and Building Materials 2022, 334, 127361, doi:10.1016/j.conbuildmat.2022.127361.
- Nguyen, M.H.; Nakarai, K.; Torrent, R. Service Life Prediction of Steam-Cured Concrete Utilizing in-Situ Air Permeability Measurements. Cement and Concrete Composites 2020, 114, 103747, doi:10.1016/j.cemconcomp.2020.103747.
- McCusker, L.B.; Von Dreele, R.B.; Cox, D.E.; Louër, D.; Scardi, P. Rietveld Refinement Guidelines. J Appl Cryst 1999, 32, 36–50, doi:10.1107/S0021889898009856.
- Finger, L.W.; Cox, D.E.; Jephcoat, A.P. A Correction for Powder Diffraction Peak Asymmetry Due to Axial Divergence. J Appl Cryst 1994, 27, 892–900, doi:10.1107/S0021889894004218.
- Li, C.; Li, Y.; Sun, H.; Li, L. The Composition of Fly Ash Glass Phase and Its Dissolution Properties Applying to Geopolymeric Materials. Journal of the American Ceramic Society 2011, 94, 1773–1778, doi:10.1111/j.1551-2916.2010.04337.x.

Reviewer 2 Report
I have following comments:
1. What is the innovation of this study? How about previous similar study? Please clarify in the Introduction part of the revised manuscript.
2. Adding granulated blast furnace slag (GBFS) to fly ash (FA) based geopolymer usually has flash setting issue. How about this study? Please clarify in the revised manuscript. You can study this issue in the literature such as: Influence of granulated blast furnace slag on the reaction, structure and properties of fly ash based geopolymer, Enhancement of the properties of fly ash based geopolymer paste by incorporating ground granulated blast furnace slag, Effect of granulated phosphorus slag on physical, mechanical and microstructural characteristics of Class F fly ash based geopolymer.
3. Granulated blast furnace slag (GBFS) can increase the strength of fly ash (FA) based geopolymer, so why you used heat curing (60±5 °C)? Please clarify in the revised manuscript.
4. Add gel phase to the SEM images.
Author Response
Reviewer #2:
- What is the innovation of this study? How about previous similar study? Please clarify in the Introduction part of the revised manuscript.
Response:
Although there have been several studies on this subject in earlier research [1–3], combining three by-products, FA, GBFS, and SF, into geopolymer mortar in Vietnam has not yet been explored, according to the reviewer's perceptive analysis of this work. The results of the current study thus point to a feasible technique for creating geopolymer mortars that support sustainable development. In other words, this research will make it possible to take a fresh approach to developing sustainable materials and resolving Vietnam's environmental issues. We have mentioned the research significance at the end of the abstract.
In addition, the revised manuscript provided more results obtained from related research in L63-69, page 2.
From:
The second approach utilises modified materials, such as silica fume (SF) and GBFS. Herein, the combination of GBFS and SF could improve the reactivity and polymerisation of FA [25–27] and is the main approach of the present study.
To:
The second approach utilises modified materials, such as silica fume (SF) and GBFS. Incorporating GBFS into FA-based geopolymers can happen differently depending on curing temperatures. At normal temperatures, the reaction is dominated by the dissolution and precipitation of C–S–H gel due to the alkali activation of GBFS. There is only a small interaction of FA and GBFS, probably due to the different kinetics of the dissolution process and species distribution. The improvement in setting time and compressive strength can be explained by forming cementitious C–S–H gel, which improved the setting and hardening of the geopolymer [25,26]. The geopolymerisation at high temperatures (e.g., 60 °C) is dominated by the combined interaction of fly ash and GBFS. This interaction is substantiated by the coexistence of C–S–H and A–S–H gel in the reaction products. The improvement in compressive strength with slag addition may be attributed to the formation of gel phases (C–S–H and A–S–H) and the compactness of microstructure [14,27,28]. In other words, the combination of GBFS and SF under high temperatures could improve the reactivity and polymerisation of FA [27,29,30] and is the main approach of the present study.
For the modifications, we have added the followings references:
[25] Li, Z.; Liu, S. Influence of Slag as Additive on Compressive Strength of Fly Ash-Based Geopolymer. Journal of Materials in Civil Engineering 2007, 19, 470–474, doi:10.1061/(ASCE)0899-1561(2007)19:6(470).
[26] Buchwald, A.; Hilbig, H.; Kaps, Ch. Alkali-Activated Metakaolin-Slag Blends—Performance and Structure in Dependence of Their Composition. J Mater Sci 2007, 42, 3024–3032, doi:10.1007/s10853-006-0525-6.
[27] Singh, B.; Ishwarya, G.; Gupta, M.; Bhattacharyya, S.K. Geopolymer Concrete: A Review of Some Recent Developments. Construction and Building Materials 2015, 85, 78–90, doi:10.1016/j.conbuildmat.2015.03.036.
[28] Zhang, P.; Gao, Z.; Wang, J.; Guo, J.; Hu, S.; Ling, Y. Properties of Fresh and Hardened Fly Ash/Slag Based Geopolymer Concrete: A Review. Journal of Cleaner Production 2020, 270, 122389, doi:10.1016/j.jclepro.2020.122389.
- Adding granulated blast furnace slag (GBFS) to fly ash (FA) based geopolymer usually has flash setting issue. How about this study? Please clarify in the revised manuscript. You can study this issue in the literature such as: Influence of granulated blast furnace slag on the reaction, structure and properties of fly ash based geopolymer, Enhancement of the properties of fly ash based geopolymer paste by incorporating ground granulated blast furnace slag, Effect of granulated phosphorus slag on physical, mechanical and microstructural characteristics of Class F fly ash based geopolymer.
Response:
We totally agree with the reviewer that flash setting issues are frequently present when adding granulated blast furnace slag (GBFS) to a fly ash (FA)-based geopolymer [4,5]. To reduce the problem, in the present study, the SR-5000F "SilkRoad" (SP) (Hanoi-Korea Co., Ltd) was used as the superplasticizer admixture (SP). In addition, adjusting the mixture proportions (by optimizing the ratio of GBFS to FA, along with activators and alkaline solutions) and controlling the water content were two other treatments used to mitigate the flash setting problem.
Based on the comment, we have clarified L155-159, page 6, in the revised manuscript.
From:
To increase the workability of the GMS mixtures, the SR-5000F "SilkRoad" (SP) (Hanoi-Korea Co., Ltd) was used as the superplasticizer admixture (SP). The specific gravity of SP was 1.12 g/сm3, confirmed by the Vietnamese Standard TCVN 8826:2011 [47]. According to the Vietnamese Standard TCVN 4506:2012, tap water (W) was utilised to prepare sodium hydroxide solution and cure FA alkali-activated mortar specimens [48].
To:
The SR-5000F "SilkRoad" (SP) was used as the superplasticizer admixture (SP). The specific gravity of SP was 1.12 g/сm3, confirmed by the Vietnamese Standard TCVN 8826:2011 [47]. According to the Vietnamese Standard TCVN 4506:2012, tap water (W) was utilised to prepare sodium hydroxide solution and cure FA alkali-activated mortar specimens [48]. Herein, the presence of SR-5000F and the minimum amount of water were designed to mitigate the flash setting problem and increase the workability of the FA-based geopolymer mortars containing GBFS [29, 49, 50].
For this modification, the following references were added:
[49] Saha, S.; Rajasekaran, C. Enhancement of the Properties of Fly Ash Based Geopolymer Paste by Incorporating Ground Granulated Blast Furnace Slag. Construction and Building Materials 2017, 146, 615–620, doi:10.1016/j.conbuildmat.2017.04.139.
[50] Wang, Y.; Xiao, R.; Hu, W.; Jiang, X.; Zhang, X.; Huang, B. Effect of Granulated Phosphorus Slag on Physical, Mechanical and Microstructural Characteristics of Class F Fly Ash Based Geopolymer. Construction and Building Materials 2021, 291, 123287, doi:10.1016/j.conbuildmat.2021.123287.
- Granulated blast furnace slag (GBFS) can increase the strength of fly ash (FA) based geopolymer, so why you used heat curing (60±5 °C)? Please clarify in the revised manuscript.
Response:
As the reviewer pointed out, Granulated blast furnace slag (GBFS) can increase the strength of fly ash (FA) based geopolymer owing to the formation of cementitious C–S–H gel. At normal temperatures, the C-S-H gel's dissolution and precipitation primarily control the calorimetric response [6,7]. Physical characteristics like setting time and the development of compressive strength at normal temperatures are mostly related to the creation of C-S-H gel as a result of slag activation by alkalis. The splitting of the peak with adding GBFS at high temperatures, i.e., 60±5 °C, shows two concurrent reaction processes, condensation of A–S–H and C–S–H. SEM and EDS experiments that verified the development of two types of reaction products, alumino-silicate gel with a Si/Al ratio of 2, and C-S-H with a Si/Al ratio of 2.5 and a Ca/Si ratio of 0.8, further corroborated this [1]. The calorimetric results show that alkali activation of GBFS is the primary reaction at normal temperatures, although both fly ash and GBFS are important contributors to polymerization at high temperatures [1–3]. Based on the findings, the authors aimed to investigate the usability of three Vietnamese by-products by using high temperatures.
Based on the comment, we have clarified L67-70, page 2, in the revised manuscript. Comment 1 of reviewer#2 was also considered in this modification.
From:
The second approach utilises modified materials, such as silica fume (SF) and GBFS. Herein, the combination of GBFS and SF could improve the reactivity and polymerisation of FA [25–27] and is the main approach of the present study.
To:
The second approach utilises modified materials, such as silica fume (SF) and GBFS. Incorporating GBFS into FA-based geopolymers can happen differently depending on curing temperatures. At normal temperatures, the reaction is dominated by the dissolution and precipitation of C–S–H gel due to the alkali activation of GBFS. There is only a small interaction of FA and GBFS, probably due to the different kinetics of the dissolution process and species distribution. The improvement in setting time and compressive strength can be explained by forming cementitious C–S–H gel, which improved the setting and hardening of the geopolymer [25,26]. The geopolymerisation at high temperatures (e.g., 60 °C) is dominated by the combined interaction of fly ash and GBFS. This interaction is substantiated by the coexistence of C–S–H and A–S–H gel in the reaction products. The improvement in compressive strength with slag addition may be attributed to the formation of gel phases (C–S–H and A–S–H) and the compactness of microstructure [14,27,28]. In other words, the combination of GBFS and SF under high temperatures could improve the reactivity and polymerisation of FA [27,29,30] and is the main approach of the present study.
In addition, we have clarified lines 197-200, page 7 in the revised manuscript.
From:
The molded samples were sealed in airtight bags and placed in a hot oven at 60±5 °C and at 70% relative humidity for six hours [54,55]. After 24 h of heat curing, all the samples were demoded and subjected to normal maintenance (temperature of 25±2 oC and humidity of approximately 90%) until testing.
To:
To promote the combined interaction of FA and GBFS, the molded samples were sealed in airtight bags and placed in a hot oven at 60±5°C and at 70% relative humidity for six hours [54,55]. After 24 h of heat curing, all the samples were demoded and subjected to normal maintenance (temperature of 25±2°C and humidity of approximately 90%) until testing. Herein, the two-stage curing cycle was used to avoid the overlapping of dissolution–precipitation reaction at ambient temperature (25±2°C) and geopolymerisation reactions at heat curing (60±5°C).
For the modifications, we have added the followings references:
[25] Li, Z.; Liu, S. Influence of Slag as Additive on Compressive Strength of Fly Ash-Based Geopolymer. Journal of Materials in Civil Engineering 2007, 19, 470–474, doi:10.1061/(ASCE)0899-1561(2007)19:6(470).
[26] Buchwald, A.; Hilbig, H.; Kaps, Ch. Alkali-Activated Metakaolin-Slag Blends—Performance and Structure in Dependence of Their Composition. J Mater Sci 2007, 42, 3024–3032, doi:10.1007/s10853-006-0525-6.
[27] Singh, B.; Ishwarya, G.; Gupta, M.; Bhattacharyya, S.K. Geopolymer Concrete: A Review of Some Recent Developments. Construction and Building Materials 2015, 85, 78–90, doi:10.1016/j.conbuildmat.2015.03.036.
[28] Zhang, P.; Gao, Z.; Wang, J.; Guo, J.; Hu, S.; Ling, Y. Properties of Fresh and Hardened Fly Ash/Slag Based Geopolymer Concrete: A Review. Journal of Cleaner Production 2020, 270, 122389, doi:10.1016/j.jclepro.2020.122389.
- Add gel phase to the SEM images.
Response:
We totally agree with the reviewer that the SEM image is one of results to show three important issues in the research object, including structure/shape of materials, change gel compositions (hydration products of Geopolymer mortar), and microstructure of mortar. Depending on the purposes of the study, researchers can use many different types of SEM images [8–10].
In this study, SEM images were used to show the structure/shape of the raw materials (Figure 2) and the microstructure of the geopolymer mortar (Figure 7).
Besides, although the authors wanted to add SEM images showing gel phases, we could not do it due to limitations in experimental conditions and samples (no material available). We have added our limitation in the conclusion part (L432-434 page 17 in the revised manuscript) for further studies. In detail, in L432-434, page 17 in the revised manuscript. Comment 10 of reviewer#2 was also considered in this revision.
From:
Given the constraints of this study, future research should analyse and evaluate the long-term evolution of compressive-flexural strength, water absorption, bulk density, and void volume.
To:
Given the constraints of this study, future research should analyse and evaluate the long-term evolution of compressive-flexural strength, bulk density, and void volume. Also, further studies should use FTIR and XRD methods to prove the newly created gels in the microstructure of mortars.

Reviewer 3 Report
In this paper, the effect of slag addition on the properties of alkali-activated materials is investigated. There are many studies on this subject in previous research. Since, the reactivity of slag is higher than that of fly ash, it is predictable to know that an increase in slag addition will enhance properties of alkali-activated materials. Therefore, the present study is not innovative enough. However, meticulous experiments were carried out in this study and therefore, it is recommended to be considered after modification.
1. L63-69, High-temperature curing is an effective method for promoting alkali-excited fly ash. The authors did not mention it. Also, what are the results of previous studies for the method proposed by the authors?
2. L73-83, I think this paragraph is redundant, the authors study alkali-activated material (slag), and this describes the role of slag in cement-based concrete.
3. L100,What are the advantages of this study compared to previous studies?
4. About literature review in the Introduction can be incorporated with more previous research.
5. Table 1, GBFS/(FA + GBFS) Change to GBFS/(FA + GBFS + SF)
6. Table 1, regarding the “Evaluated properties”, should add information on the test time, specimen type, etc., for each experiment.
7. The methods and standards by which the physical information in Table 2 was measured need to be described, especially the glass phase content.
8. L272, It should be interpreted as: the particle size of slag is between fly ash and silica fume, .....
9. Figure 7sem image is too blurred. It should be replaced.
10. L290-293, As the authors raise, slag has a high CaO and Al2O3 content. Therefore, CASH is generated, not NASH. The authors should prove it by FTIR and XRD.
11. L304, Explain what "extra hydration products" refers to.
The English language is suitable for scientific article.
Author Response
Reviewer #3:
In this paper, the effect of slag addition on the properties of alkali-activated materials is investigated. There are many studies on this subject in previous research. Since, the reactivity of slag is higher than that of fly ash, it is predictable to know that an increase in slag addition will enhance properties of alkali-activated materials. Therefore, the present study is not innovative enough. However, meticulous experiments were carried out in this study and therefore, it is recommended to be considered after modification.
Response:
Thank you very much for your insightful analysis.
- L63-69, High-temperature curing is an effective method for promoting alkali-excited fly ash. The authors did not mention it. Also, what are the results of previous studies for the method proposed by the authors?
Response:
Based on the comment, we have updated L63-69, page 2, in the revised manuscript with related studies. Comment 3 of reviewer#1 and comment 4 of reviewer#2 were also considered in this modification.
From:
The second approach utilises modified materials, such as silica fume (SF) and GBFS. Herein, the combination of GBFS and SF could improve the reactivity and polymerisation of FA [25–27] and is the main approach of the present study.
To:
The second approach utilises modified materials, such as silica fume (SF) and GBFS. Incorporating GBFS into FA-based geopolymers can happen differently depending on curing temperatures. At normal temperatures, the reaction is dominated by the dissolution and precipitation of C–S–H gel due to the alkali activation of GBFS. There is only a small interaction of FA and GBFS, probably due to the different kinetics of the dissolution process and species distribution. The improvement in setting time and compressive strength can be explained by forming cementitious C–S–H gel, which improved the setting and hardening of the geopolymer [25,26]. The geopolymerisation at high temperatures (e.g., 60 °C) is dominated by the combined interaction of fly ash and GBFS. This interaction is substantiated by the coexistence of C–S–H and A–S–H gel in the reaction products. The improvement in compressive strength with slag addition may be attributed to the formation of gel phases (C–S–H and A–S–H) and the compactness of microstructure [14,27,28]. In other words, the combination of GBFS and SF under high temperatures could improve the reactivity and polymerisation of FA [27,29,30] and is the main approach of the present study.
In addition, we have clarified L197-200, page 7, in the revised manuscript.
From:
The molded samples were sealed in airtight bags and placed in a hot oven at 60±5 °C and at 70% relative humidity for six hours [54,55]. After 24 h of heat curing, all the samples were demoded and subjected to normal maintenance (temperature of 25±2 oC and humidity of approximately 90%) until testing.
To:
To promote the combined interaction of FA and GBFS, the molded samples were sealed in airtight bags and placed in a hot oven at 60±5°C and at 70% relative humidity for six hours [54,55]. After 24 h of heat curing, all the samples were demoded and subjected to normal maintenance (temperature of 25±2°C and humidity of approximately 90%) until testing. Herein, the two-stage curing cycle was used to avoid the overlapping of dissolution–precipitation reaction at ambient temperature (25±2°C) and geopolymerisation reactions at heat curing (60±5°C).
For the modifications, we have added the followings references:
[25] Li, Z.; Liu, S. Influence of Slag as Additive on Compressive Strength of Fly Ash-Based Geopolymer. Journal of Materials in Civil Engineering 2007, 19, 470–474, doi:10.1061/(ASCE)0899-1561(2007)19:6(470).
[26] Buchwald, A.; Hilbig, H.; Kaps, Ch. Alkali-Activated Metakaolin-Slag Blends—Performance and Structure in Dependence of Their Composition. J Mater Sci 2007, 42, 3024–3032, doi:10.1007/s10853-006-0525-6.
[27] Singh, B.; Ishwarya, G.; Gupta, M.; Bhattacharyya, S.K. Geopolymer Concrete: A Review of Some Recent Developments. Construction and Building Materials 2015, 85, 78–90, doi:10.1016/j.conbuildmat.2015.03.036.
[28] Zhang, P.; Gao, Z.; Wang, J.; Guo, J.; Hu, S.; Ling, Y. Properties of Fresh and Hardened Fly Ash/Slag Based Geopolymer Concrete: A Review. Journal of Cleaner Production 2020, 270, 122389, doi:10.1016/j.jclepro.2020.122389.
- L73-83, I think this paragraph is redundant, the authors study alkali-activated material (slag), and this describes the role of slag in cement-based concrete.
Response:
As the reviewer commented, lines 73-83 show the role of GBFS in cement based-concrete via several notable previous research. In addition, the role of SF in the system was also highlighted via previous studies (L78-83, page 2).
Based on the comment, we have condensed the content lines 70-78, page 2, in the revised manuscript to prevent redundancy.
From:
When applied to alkali-activated materials, GBFS is a precursor for aluminosili-cates rich in calcium. For example, important material characteristics such as com-pressive strength, density, and sulfate resistance may be enhanced by the addition of GGBFS [28,29]. In cement, the inclusion of GBFS may reduce porosity and enhance corrosion resistance [30,31]. Additionally, it has been demonstrated that GBFS en-hances the sulfate resistance of concrete by reducing the C3A concentration and soluble Ca(OH)2 [32–34]. Higgins evaluated the effect of the GBFS replacement ratio on the sulfate resistance in concrete and found that the GBFS concentration and sulfate re-sistance were positively associated [34,35]. SF is a by-product of ferrosilicon production. It has an extremely fine particle size and is mostly used as a filler. It is a highly reactive pozzolanic material because it contains significant amorphous silicon dioxide. In addition to increasing the strength, the use of SF improves the cohesion between the binder and aggregates owing to its tiny particle size. Thus, reducing the mixture's bleeding rate and segregation potential in its fresh form [36,37].
To:
When applied to alkali-activated materials, GBFS acts as a precursor for calcium-rich aluminosilicates, enhancing properties like compressive strength, density, and sulfate resistance [31,32]. Its inclusion in cement can reduce porosity and improve corrosion resistance. Studies have shown that GBFS positively affects sulfate resistance in concrete by reducing C3A concentration and soluble Ca(OH)2, with the concentration of GBFS and sulfate resistance being positively correlated [33,34]. SF is a by-product of ferrosilicon produc-tion. It has an extremely fine particle size and is mostly used as a filler. It is a highly reactive pozzolanic material because it contains significant amorphous silicon dioxide. In addition to increasing the strength, the use of SF improves the cohesion between the binder and aggregates owing to its tiny particle size. Thus, reducing the mixture's bleeding rate and segregation potential in its fresh form [35,36].
- L100,What are the advantages of this study compared to previous studies?
Response:
As the insightful analysis of the reviewer on this study, although there are several studies on this topic in previous research [1–3], in Vietnam, incorporating three by-products FA, GBFS, and SF, into geopolymer mortar has not been investigated yet. Thus, the present research findings reveal a promising method for producing geopolymer mortars that promote sustainable development. In other words, this research will open up a new approach to creating sustainable materials and solving environmental problems in Vietnam.
- About literature review in the Introduction can be incorporated with more previous research.
Response:
Based on the comment, we have updated L63-69, page 2, in the revised manuscript with related studies. Comment 3 of reviewer#1 and comment 4 of reviewer#2 were also considered in this modification.
From:
The second approach utilises modified materials, such as silica fume (SF) and GBFS. Herein, the combination of GBFS and SF could improve the reactivity and polymerisation of FA [25–27] and is the main approach of the present study.
To:
The second approach utilises modified materials, such as silica fume (SF) and GBFS. Incorporating GBFS into FA-based geopolymers can happen differently depending on curing temperatures. At normal temperatures, the reaction is dominated by the dissolution and precipitation of C–S–H gel due to the alkali activation of GBFS. There is only a small interaction of FA and GBFS, probably due to the different kinetics of the dissolution process and species distribution. The improvement in setting time and compressive strength can be explained by forming cementitious C–S–H gel, which improved the setting and hardening of the geopolymer [25,26]. The geopolymerisation at high temperatures (e.g., 60 °C) is dominated by the combined interaction of fly ash and GBFS. This interaction is substantiated by the coexistence of C–S–H and A–S–H gel in the reaction products. The improvement in compressive strength with slag addition may be attributed to the formation of gel phases (C–S–H and A–S–H) and the compactness of microstructure [14,27,28]. In other words, the combination of GBFS and SF under high temperatures could improve the reactivity and polymerisation of FA [27,29,30] and is the main approach of the present study.
For the modifications, we have added the followings references:
[25] Li, Z.; Liu, S. Influence of Slag as Additive on Compressive Strength of Fly Ash-Based Geopolymer. Journal of Materials in Civil Engineering 2007, 19, 470–474, doi:10.1061/(ASCE)0899-1561(2007)19:6(470).
[26] Buchwald, A.; Hilbig, H.; Kaps, Ch. Alkali-Activated Metakaolin-Slag Blends—Performance and Structure in Dependence of Their Composition. J Mater Sci 2007, 42, 3024–3032, doi:10.1007/s10853-006-0525-6.
[27] Singh, B.; Ishwarya, G.; Gupta, M.; Bhattacharyya, S.K. Geopolymer Concrete: A Review of Some Recent Developments. Construction and Building Materials 2015, 85, 78–90, doi:10.1016/j.conbuildmat.2015.03.036.
[28] Zhang, P.; Gao, Z.; Wang, J.; Guo, J.; Hu, S.; Ling, Y. Properties of Fresh and Hardened Fly Ash/Slag Based Geopolymer Concrete: A Review. Journal of Cleaner Production 2020, 270, 122389, doi:10.1016/j.jclepro.2020.122389.
- Table 1, GBFS/(FA + GBFS) Change to GBFS/(FA + GBFS + SF)
Response:
Based on the comment, we have updated L119, page 3, in the revised manuscript.
From:
GBFS/(FA + GBFS), (wt.%)
To:
GBFS/(FA + GBFS + SF), (wt.%)
- Table 1, regarding the “Evaluated properties”, should add information on the test time, specimen type, etc., for each experiment.
Response:
At the beginning of the experimental program part, we would like to present an outline of the experimental variable via Table 1. This approach is commonly used in previous research [11–13]. Then, the detailed contents, including experimental time, experimental age, specimen type, size, and number of samples, were presented in the following sections from 2.2 to 2.5. Please kindly refer to the corresponding sections for more information.
- The methods and standards by which the physical information in Table 2 was measured need to be described, especially the glass phase content.
Response:
Figure 3 illustrates the X-ray diffraction (XRD) patterns of FA, GBFS, and FS-90 created by a D2-Phaser X-ray diffractometer using Cu-K radiation. Through Rietveld refining, the signals for crystalline phases, including mullite, quartz, and calcium oxide, can also be discovered [14,15]. For instance, a peak of 2-theta angle spanning from 16o to 28o was found for the GGBFS, an amorphous hump primarily made up of a glassy phase. Due to the presence of unburned carbon (Loss on ignition), Eq. (1) was used to determine the glass phase composition [16], as listed in Table 2.
Glass phase = 100 - mullite - quartz - calcium oxide - loss on ignition (1)
Since it is a commonly used method, we would like to cite related papers in the revised manuscript rather than a detailed description. Based on the comment, we have updated related information in L123-124, page 4, in the revised manuscript.
From:
Table 2 displays the X-ray fluorescence analysis-detected physical characteristics and chemical compositions of FA, GBFS, and SF-90.
To:
Table 2 displays the physical characteristics and chemical compositions of FA, GBFS, and SF-90 using the Rietveld refinement with XRD [47-49] and the X-ray fluorescence analysis, respectively.
For the modifications, we have added the followings references:
[47] McCusker, L.B.; Von Dreele, R.B.; Cox, D.E.; Louër, D.; Scardi, P. Rietveld Refinement Guidelines. J Appl Cryst 1999, 32, 36–50, doi:10.1107/S0021889898009856.
[48] Finger, L.W.; Cox, D.E.; Jephcoat, A.P. A Correction for Powder Diffraction Peak Asymmetry Due to Axial Divergence. J Appl Cryst 1994, 27, 892–900, doi:10.1107/S0021889894004218.
[49] Li, C.; Li, Y.; Sun, H.; Li, L. The Composition of Fly Ash Glass Phase and Its Dissolution Properties Applying to Geopolymeric Materials. Journal of the American Ceramic Society 2011, 94, 1773–1778, doi:10.1111/j.1551-2916.2010.04337.x.
- L272, It should be interpreted as: the particle size of slag is between fly ash and silica fume, .....
Response:
Based on the comment, we have updated L272, page 10, in the revised manuscript.
From:
In addition, the smaller particle size of GBFS compared to that of FA could be another reason.
To:
In addition, the particle size of GBFS is between FA and SF could be another reason.
- Figure 7sem image is too blurred. It should be replaced.
Response:
Based on the comment, we have replaced the better SEM images with high resolution in Figs 2 and 7.
- L290-293, As the authors raise, slag has a high CaO and Al2O3 content. Therefore, CASH is generated, not NASH. The authors should prove it by FTIR and XRD.
Response:
Based on the comment, we have updated L290-293, page 10, in the revised manuscript.
From:
It improves the microstructure of geopolymer mortars by generating Na2O-Al2O3-SiO2-H2O, CaO-SiO2-H2O, and Al2O3-SiO2-H2O gels.
To:
It improves the microstructure of geopolymer mortars by generating CaO- Al2O3-SiO2-H2O, CaO-SiO2-H2O, and Al2O3-SiO2-H2O gels.
As the reviewer suggested, providing FTIR and XRD results of GBFS is an effective way to support the discussion points. Unfortunately, the authors could not reach these additional results due to the limitation of experimental equipment and the lack of samples at this revised time. We are sorry for this limitation.
Based on the comment, we have added our limitation in the conclusion part (L432-434 page 17 in the revised manuscript) for further studies. In detail, in L432-434, page 17 in the revised manuscript.
From:
Given the constraints of this study, future research should analyse and evaluate the long-term evolution of compressive-flexural strength, water absorption, bulk density, and void volume.
To:
Given the constraints of this study, future research should analyse and evaluate the long-term evolution of compressive-flexural strength, bulk density, and void volume. Also, further studies should use FTIR and XRD methods to prove the newly created gels in the microstructure of mortars.
- L304, Explain what "extra hydration products" refers to.
Response:
Based on the comment, we have provided more information in L303-305, page 11 in the revised manuscript.
From:
The increase in compressive strength may result from the formation of extra hydration products, which fill the porous microstructure of the geopolymer mortar [62,63].
To:
The increase in compressive strength may result from the formation of extra hydration products by generating CaO-Al2O3-SiO2-H2O, CaO-SiO2-H2O, and Al2O3-SiO2-H2O gels, which fill the porous microstructure of the geopolymer mortar [62,63]

Round 2
Reviewer 2 Report
Publish
Author Response
Thank you very much for your positive evaluation.